# “Now I Feel a Little Bit More Secure”: The Impact of SNAP Enrollment on Older Adult SSI Recipients

**DOI:** 10.3390/nu13124362

**Published:** 2021-12-04

**Authors:** Katie Savin, Alena Morales, Ronli Levi, Dora Alvarez, Hilary Seligman

**Affiliations:** 1School of Social Welfare, University of California, Berkeley, CA 94709, USA; 2MSW Program, School of Health Sciences, University of the Pacific, Sacramento, CA 95817, USA; 3Department of Nutritional Sciences and Toxicology, University of California, Berkeley, CA 94709, USA; alenamorales@berkeley.edu; 4Center for Vulnerable Populations, University of California, San Francisco, CA 94143, USA; ronli.levi@ucsf.edu (R.L.); Hilary.Seligman@ucsf.edu (H.S.); 5School of Medicine, University of California, San Francisco, CA 94143, USA; dshirly@stanford.edu; 6School of Medicine, Stanford University, Stanford, CA 94305, USA

**Keywords:** SNAP, food insecurity, poverty, social determinants of health, qualitative research

## Abstract

In June 2019, California expanded Supplemental Nutrition Assistance Program (SNAP) eligibility to Supplemental Security Income (SSI) beneficiaries for the first time. This research assesses the experience and impact of new SNAP enrollment among older adult SSI recipients, a population characterized by social and economic precarity. We conducted semi-structured, in-depth interviews with 20 SNAP participants to explore their experiences with new SNAP benefits. Following initial coding, member-check groups allowed for participants to provide feedback on preliminary data analysis. Findings demonstrate that SNAP enrollment improved participants’ access to nutritious foods of their choice, contributed to overall budgets, eased mental distress resulting from poverty, and reduced labor spent accessing food. For some participants, SNAP benefit amounts were too low to make any noticeable impact. For many participants, SNAP receipt was associated with stigma, which some considered to be a social “cost” of poverty. Increased benefit may be derived from pairing SNAP with other public benefits. Together, the impacts of and barriers to effective use of SNAP benefits gleaned from this study deepen our understanding of individual- and neighborhood-level factors driving health inequities among low-income, disabled people experiencing food insecurity and SNAP recipients.

## 1. Introduction

Food insecurity refers to the limited or uncertain ability to reliably access safe and nutritious food. In 2019, more than 13.7 million US households were food insecure. Approximately 21% of food insecure households included an adult over the age of 65 years [1]. Food insecurity disproportionately affects marginalized older adults such as those who are disabled, lower income, lesbian, gay, bisexual and/or transgender, and Black or Latino [2,3,4]. 

Food insecurity is linked with poor dietary intake, adverse mental and physical health, and increased health care utilization and expenditures [5,6,7]. Among older adults, food insecurity is associated with increased fall risk and more severe depression, heart failure, asthma, osteoporosis, and cognitive impairment [8,9]. Food insecure older adults are also more likely to experience cost-related medication non-adherence, further increasing their risk of poor health outcomes [10].

The federal Supplemental Nutrition Assistance Program (SNAP) provides 12.3% of Americans with money to buy food [11]. SNAP is the nation’s most effective and comprehensive strategy for reducing food insecurity [12,13]. Most rigorous studies of SNAP participation demonstrate that it supports improved diet quality [14], reductions in BMI [14], and better health (including among those with diet-sensitive chronic disease such as diabetes) [15,16,17]. SNAP participation is also associated with lower annual health care expenditures (−$1409) among adults, especially for those with hypertension, coronary heart disease, or disability [3]. Older adults dually eligible for Medicare and Medicaid (federal health insurance programs for the elderly, disabled, or poor) are less likely to be admitted to a hospital or nursing home (and when admitted, have a shorter length of stay) if they are enrolled in SNAP. In addition, SNAP participation may reduce cost-related medication non-adherence [18]. 

Historically, Supplemental Security Income (SSI) recipients in California have not been eligible to receive SNAP benefits, despite their monthly benefit levels placing them within SNAP criteria for income eligibility. SSI is a federal program providing monthly cash payments to low-income adults who are aged 65 years and older, blind, or disabled. For the purposes of SSI, disability is determined by a formal assessment of whether applicants’ impairments render them incapable of engaging in the workforce based on their job skills, employment history and education [19]. Thus, the nature and severity of disability can vary amongst recipients. 

SSI beneficiaries are a uniquely vulnerable group due to restrictions on work earnings, low benefit levels (an average of $794 monthly), and limitations on allowed assets to $2000 [19]. Each of these policies increase need to rely on public benefits to meet basic needs. Among disability assistance program recipients, SSI beneficiaries experience the highest rates of food insecurity, likely due to the high costs of disability coupled with inadequate SSI benefit levels [20,21]. 

The roots of SSI beneficiaries’ exclusion from SNAP in California lie in the inception of SSI in 1974, when states were allowed a “cash-out” option to ease administrative costs. This option allowed states to include an SSI recipient’s SNAP allotment, calculated at $10 per person, in their SSI state supplement and eliminate their eligibility for additionally enrolling in SNAP. Yet, the SNAP allotment within SSI benefits was never formally increased, while the exclusion from SNAP benefits persisted in California [22,23]. Because of reductions in purchasing power in the ensuing decades, many SSI recipients in California have been left with inadequate funds to purchase basic needs such as food [24]. 

In June 2019, California became the last state in the US to eliminate its cash-out policy for SSI beneficiaries (California State Assembly Bill 1811), thus expanding SNAP eligibility to California’s SSI population for the first time. Using a mixed-methods approach, we sought to examine the ways in which new SNAP eligibility and subsequent enrollment impacts older adult dietary intake, stress, and health. The results of the quantitative study, published elsewhere [25], revealed high SNAP uptake in San Francisco after the policy change (73%) and reductions in food security. However, there were no observable differences in other outcomes between SNAP and non-SNAP recipients after the policy change. Qualitative interviews can help add nuance and context to these quantitative results. To our knowledge, this is the first study to qualitatively examine the impact of SNAP among an older adult cohort of newly eligible SNAP beneficiaries.

## 2. Materials and Methods

This qualitative study was conducted in tandem with a quantitative component designed to assess the impact of new SNAP enrollment on dietary outcomes and food insecurity among a cohort of SSI recipients (*n* = 213) in San Francisco. Between November 2019 and February 2020, we conducted in-depth, one on one qualitative interviews, lasting 60–90 min, among a subset (*n* = 20) of survey participants (see Table 1) who indicated interest during the quantitative component. Inclusion criteria for the quantitative component included receipt of SSI benefits, age greater than 17 years, ability to complete a survey and dietary recalls in English, and not receiving SNAP benefits at baseline. Participants were recruited from low-income housing sites, congregate meal sites, SNAP outreach events, and via flyers and word of mouth. Inclusion criteria for the qualitative component included prior participation in the survey, age greater than 49 years, and new SNAP enrollment since the policy change.

Two trained study personnel interviewed participants using a semi-structured interview guide (see Appendix A). Interview questions explored participants’ perceptions of SNAP and its relationship to their health and dietary intake. The interview format allowed flexibility to explore topics introduced by the study participant. In August 2020, following initial stages of data analysis, participants were invited to participate via telephone or Zoom, a video conferencing platform, in one of three feedback sessions. During these sessions, researchers shared preliminary findings and elicited feedback. The goal of the feedback sessions was to ensure fidelity to research participants’ experiences in the data analysis [26]. Participants were compensated with a $30 gift card for each portion of the study. 

Study procedures were approved by the University of California, San Francisco’s Institutional Review Board. Verbal consent was obtained and recorded for all study participants.

Interviews were digitally recorded and transcribed. Participant characteristics, including age, race/ethnicity, and SNAP benefit amount, were linked to each transcript. Analysis was facilitated by Dedoose Version 9.0.3 (SocioCultural Research Consultants, Manhattan Beach, USA) qualitative data analysis software. 

Transcripts were analyzed using Braun and Clarke’s six-step framework for thematic analysis of qualitative data [27]. The first step, familiarizing oneself with the data, was accomplished in the process of listening to the audio recordings and editing transcripts to prepare them for analysis. Researchers individually completed the second step, generating initial codes, during their first reading of the transcripts. Both semantic and latent approaches to coding were used. Since the same questions were asked of all participants, some of the data fell neatly into predetermined categories, from which a set of semantic, or overt, codes and child (or sub) codes were developed. At the same time, the interpretive lens of the food insecurity and related health outcomes literature was used for latent, or covert, coding in order to capture the experiences, subtler meanings, and stigmas that participants expressed in relation to their experiences with SNAP. All initial, unique codes from each coder’s list were compiled into a codebook. 

The third step involved searching for themes. The parent and child code function in Dedoose was used to group similar codes together and identify wider organizing concepts. In step four, the team reviewed potential themes, producing a natural transition into the fifth step: defining and naming. At this step, new codes were identified, some existing codes were collapsed, and some themes were renamed and reorganized [27]. 

The second through fifth steps were conducted as a team of three coders in both individual and group formats. A series of four, hour-long team meetings allowed the team to discuss points of convergence and departure in coding, to conceptualize an organizing framework for themes in the data, and to raise concerns around reflexivity and potential bias in our individual work. Ultimately, five core themes were identified for inclusion in this manuscript. A table with the five themes, the root and child codes they emerged from, and code definitions is included in Appendix B.

The team-coding approach enhances the rigor of qualitative analysis. Through collaboration amongst three coders, multiple disciplinary and personal perspectives were introduced to the coding process. The team consisted of one social welfare PhD candidate, one medical intern, and one dietetics student. Two team members identified openly as disabled and all three had personal experience receiving either SNAP or disability benefits. Team meetings served as opportunities for reflexivity, as coders had to make explicit their coding processes in order to communicate them to each other. Lastly, team coding served as analytic triangulation which challenged implicit biases and strengthened the ultimate findings. An audit trail was kept to document the coding process and enhance rigor [28]. These components, coupled with the feedback sessions, contribute to trustworthiness of the study’s findings.

## 3. Results

Five themes emerged from thematic analysis: the impact of SNAP benefits on participants’ lives; interactions between optimal SNAP usage and other social programs (e.g., housing); labor required to access food resources; the role of food beyond nutrition; and stigma associated with SNAP receipt. Except where noted, all quotes were derived from in-depth, one on one interviews.

### 3.1. Impact of SNAP

Participants reported being able to buy more adequate amounts of food (“But now at least we get through the month”) as well as more nutritious food (“Oh, yeah, I’ve been getting more healthy food since I’ve had [SNAP]”) since they had enrolled in SNAP. For some participants, SNAP enrollment provided access to previously unaffordable fresh produce. Other foods that participants reported being able to buy as a result of their SNAP benefits were items such as prepared salads and hot meals (Despite federal policy prohibiting the purchase of prepared foods with SNAP benefits, some participants in our study were able to use their SNAP benefits to buy hot meals through a county-run SNAP Restaurant Meals Program (RMP). The RMP allows approved SNAP recipients who are older adults, experiencing homelessness, or disabled, and therefore with limited ability to prepare food from raw ingredients, to purchase prepared meals at designated restaurants using their SNAP benefits). Since many participants struggled with food preparation due to lack of available kitchen space or disability, being able to purchase nutritious, prepared food was an important benefit.

The addition of SNAP funds allowed participants to shift away from coping mechanisms for food insecurity that they disliked or came with added burdens. For example, participants described SNAP benefits as allowing them to stop borrowing money from friends, going without fresh foods, and depending on community resources such as free dining rooms. Several participants were accustomed to running out of money to buy food well before the end of the month and depending on congregate meal sites and other community resources for their food. While eating at congregate meal sites could provide opportunities for community building, they also involved an investment of time and energy to access and conditions that were not always pleasant, such as bad odors and loud environments. The ability to purchase food and not have to depend solely on these community resources provided “stress relief”, “freedom”, and “a feeling of security”. 

For many participants, the impact of SNAP was at once material and psychological, providing the security of food and peace of mind. One participant explained, “When I first got [SNAP], I used it to stock up on tomato sauce and the things that I usually don’t have in the house. So now I feel a little bit more secure that I can always make something even if I don’t have any meat in the refrigerator. I can figure out something”. This sense of security extended beyond food purchases, as the addition of SNAP funds to participants’ overall budgets freed up money to spend on other necessities, including shoes, underwear, bedding, toiletries, toilet paper, laundry detergent, clothes, kitchenware, and the occasional movie ticket. The nature of the items that participants spent their newly available money on, primarily basic necessities, is consistent with ongoing material need. Participants also noted that the extra funds allowed them to avoid low-balance and overdraft banking fees, pay back debts and overdue bills, and stay current on new bills. 

Our findings affirm that SSI benefits were not adequate to cover food needs prior to the policy change. As one participant reflected; “It’s a little bit like robbing Peter to pay Paul, and maybe I can pay $30 on my PG&E bill…but nothing is ever like you’re ahead”. Here, they explained that as a result of their SNAP benefit, they were able to free up money that was immediately used to pay a PG&E (electric) bill, suggesting that they routinely struggled to make ends meet. 

Some participants received very low SNAP benefit amounts and thus did not experience an appreciable impact from their new benefits. Among our 20 participants, eight received less than $50 per month, the inadequacy of which was compounded by the high cost of living in San Francisco. As a participant explained, “because food is so expensive, it hasn’t really, really, really made a difference. Like I said, if it do make a difference, I’d have to go way across town to find a bargain.” Since the time of these interviews, response to the COVID-19 pandemic has included an increase in benefit levels for all SNAP recipients to the maximum allowable based on the number of people in a household, which is $194 for an individual.

### 3.2. Interaction of Benefits

Receipt of other public benefits in addition to SNAP and SSI augmented participants’ ability to make use of SNAP. This was particularly true with the California Home and Community-Based Services (HCBS) program, In-Home Supportive Services (IHSS) and housing-related benefits such as the federal housing subsidy known as Section 8 and public housing. IHSS provides payment to a caregiver to come to the home and assist with instrumental activities of daily living, including grocery shopping and meal preparation. It is designed to support the needs of older or disabled people living in the community and prevent the need for institutionalization. Three interview participants discussed current IHSS benefits (though none were directly asked) and one discussed a recently submitted IHSS application. 

Some participants who did not receive IHSS cited the physical and mental challenges of grocery shopping and meal preparation as barriers to their nutritional intake, limiting the impact of their SNAP benefits. For example, one participant described an access–affordability dichotomy arising from her osteoarthritis diagnosis and neighborhood grocery availability. The stores nearest to their home sold prepared foods that were either expensive or unhealthy: “Then there’s [my neighborhood pizza restaurant] … the cheese is way too salty and it’s not good for me. I have high blood pressure. I told them, “No cheese on nothing.” It hurts my stomach and makes my blood pressure rise… Then there’s the little deli store across the street... To buy a little cup of fruit salad may cost me $4, $5. You pay for convenience”. This participant, like many others, preferred to get groceries at the larger and less expensive grocery store across town, however that required more walking and a bus trip. They described a recent trip: “It was quite a struggle. By the time I get home and unpack everything, I’m wore out. I don’t want to cook. I don’t want to do nothing but lay down.” At the time of the interview, they had an appointment with a county social worker to determine their eligibility for IHSS. IHSS would allow them to use SNAP funds at less expensive stores that better served their dietary needs.

Due to the high cost of housing in San Francisco, most SSI beneficiaries cannot afford market-rate housing and therefore depend on housing benefits. Among our twenty participants, five had Section 8 housing, eight lived in a single-room occupancy (SRO) hotel room, five lived in supportive housing with low-income subsidies, and one lived in senior housing. The majority of our participants lived alone and had limited space for food storage. Participants described the direct and significant effects their housing had on their ability to prepare nutritious meals. Those living in SRO’s did not have typical kitchen appliances; instead, their rooms were generally equipped with a mini fridge, a microwave, a sink and limited food storage space. Buying prepared food, which would not require as much kitchen equipment, was typically both prohibitively expensive and ill-suited to participants’ dietary restrictions. Participants described adaptations including chopping food while sitting on the bed, using electric appliances such as burner plates, crockpots, and electric skillets, and purchasing microwavable food bowls that serve as complete meals. 

Many hotels and housing units had communal kitchens instead of (or in addition to) these small in-room kitchens. Barriers to using communal kitchens included being occupied by neighbors, physical barriers to carrying food back and forth from one’s room to the kitchen, and food theft. 

Participants without adequate access to public benefits had to rely on community resources. However, several participants described associated challenges here as well, including receiving food from the food pantry that was on the verge of spoiling and free groceries that were often in large quantities of a single type, which was difficult to consume and store. One participant gave an account of their access to perishable groceries such as fresh produce and meat: “I get plenty of that from the food pantry, but a lot of it I give away. Cause I don’t have much storage space, I’m going to give them to my friend in the building.” Sharing food with neighbors was a solution that came up frequently during interviews and allowed for community building and mutual support. 

### 3.3. Labor for Food

While many participants reported that SNAP has allowed more flexibility in their grocery budget, labor to access cheap, adequate nutrition was widely reported as both time-consuming and draining. Safety also limited a few participants’ ability to shop at certain places or at certain points in the day. One participant described the labor required to take a simple trip to the grocery store: “It takes time, energy, and it’s a real hassle because [grocery store] is near the [location]. That’s a long ways from home. That’s into another neighborhood. And that’s not a good neighborhood, because it’s a really high crime area… so, I try to go there during the daytime and get out.”

The inconvenient location of food resources created the need for additional labor and reduced food purchasing power. As previously mentioned, several participants lived in SRO’s with inadequate kitchen space or refrigeration, limiting their ability to purchase produce in quantities greater than those lasting a day or two. In these cases, participants settled for convenience meals and ready-made salads that oftentimes cost more than raw ingredients. Others described how the built environment exacerbated their disability or chronic illness, limiting their food access. For example, lack of elevator access in her housing unit frustrated one participant who shops alone: “It was like a chore to go to the grocery store, and I lived up 28 steps, so I would have to carry the groceries by myself.” Most participants also did not have a car, so shopping trips became all-day excursions that often had to be repeated frequently, as they could not carry adequate groceries to last for long durations. Traveling by bus for several hours at a time sometimes led to increased chronic pain, or general exhaustion caused by exacerbation of disability. “I was having a back problem… [the bus] is always crowded so I have to stand,” one participant explained. Corner stores and restaurants accessible by foot were often noted to have limited variety and higher prices.

Other available food benefits were also not able to overcome these barriers. For example, the Market Match program allows SNAP recipients to double their benefit dollars used to purchase fresh, locally grown produce at farmers’ markets. While Market Match lessened the financial burden of food and encouraged the consumption of produce, older adults experienced additional barriers (such as traveling to a different neighborhood) that limited their capacity to access these resources and, in some cases, made them unusable. In contrast, Meals on Wheels, though only available to adults 65 and older, proved to be a convenient complement to SNAP for two participants who were experiencing transportation barriers. One of these two participants noted that SNAP provides more freedom of choice, but Meals on Wheels was less labor intensive when they were experiencing acute medical issues. 

For several participants, SNAP reduced the amount of labor needed to access food. Some participants reported buying groceries to prepare at home with SNAP benefits instead of having to go to congregate meal sites for the majority of their meals in the second half of each month. This impact was particularly important for SSI recipients, for whom excess travel was often a significant barrier.

### 3.4. Role of Food

In addition to providing sustenance, food was described as playing an important role in culture, health, and society. Multiple participants noted food preferences and cooking strategies were culturally significant because of the relationship to their childhood. One participant noted that the food budget increase provided by SNAP allowed him to purchase nostalgic foods: “I’m Portuguese, so growing up we ate a lot of linguica…So, I thought oh, but that’s expensive. It was like 7 or 8 dollars. And I thought oh I’m going to get this. I’ve never been able to buy linguica and I love it… So, I’m in a position where I can get a few special things that I’ve never been able to get before.” Culturally significant foods were also noted to be vital community-building strategies, allowing participants to purchase special foods for socializing with family or friends. 

Participants also built community through the shared struggle for food adequacy by sharing extra food with neighbors. While most participants reported that SNAP benefits reduced their need to visit food pantries or senior meal centers, a few continued to attend in order to socialize with neighborhood friends. One participant explained their ongoing senior center attendance as fulfilling a social need separate from nutrition: “It’s really also social interaction as well, because it provides an opportunity to see people that I know. It’s an opportunity because as you’re probably aware seniors have a tendency to isolate.”

### 3.5. Stigma

Stigma was never explicitly raised by interviewers. However, almost all participants referenced the stigmatizing experience of receiving public benefits, “food stamps” in particular (prior to electronic benefits transfer (EBT) cards, SNAP payments were distributed via paper vouchers known as food stamps, a term which remains a common colloquialism for SNAP benefits). For some, this reference seemed to indicate an effort to distinguish themselves from the stigma associated with food stamps by assigning that stigma to other recipients, and then contrasting their SNAP usage as morally superior. This was done alternately by referencing fraudulent uses of SNAP and even recommending more stringent fraud prevention policies, emphasizing the health benefits of the foods they purchased with SNAP funds, or highlighting their ascetic lifestyle by discussing how little money or food they needed compared to others. One participant combined these strategies: “But for me you know, I really don’t need a lot of money. When it’s just you and you don’t smoke, drink, or do drugs then your money can go for the right things. I’ve always believed that if you live your life the right way you really don’t need a lot of money.” 

Each participant was asked to provide input on the policy change that extended SNAP benefits to them. Some responded that SNAP should have stricter rules in an attempt to put restrictions on how they perceived other SNAP recipients were using, or misusing, their benefits. One participant sought to protect SNAP from fraud by recommending that EBT be banned at all corner stores. A few minutes after disclosing that they had recently stopped smoking, this participant said, “Yeah, but what’s hurting the EBT program is these corner stores falsifying records that the person is buying groceries when they’re getting beer, cigarettes, or whatever they need, or hard alcohol.” Another participant had even more stringent policy recommendations that they related to concerns over people purchasing unhealthy food. They suggested that SNAP should only be used to purchase fresh fruits and vegetables, and that for everything else, people should “take your [own] money and buy snacks.”

Because stigma emerged as a prominent theme but was not directly addressed in the interview guide, the authors opted to address it directly during the group feedback sessions. When we suggested to participants that they might experience stigma or negative judgement from SNAP utilization, some participants skipped the question entirely. Others were emphatic that they did not care what people thought about them, and “who are they to judge?” suggesting that they did experience stigma. Another participant indicated that young people might be concerned about stigma, “but when you get older, that’s not a concern. You’re trying to survive, that’s the only thing you care about.” Still others spoke poignantly about their struggles living in poverty in a high-cost city like San Francisco: “The aspect of dignity is less valuable than survival skills. In general, being poor is a very humbling and sometimes humiliating experience. I find stigma in people who are poor… I’m not gonna mince words: It sucks being poor.” Other group members echoed this participant’s sentiments and agreed that survival at times came at the cost of dignity. 

Yet participants also made clear that SNAP policy could limit the extent of this dignity cost. In reference to the switch from paper food stamps to the EBT card, one participant opined that the card was “more humanizing… less humiliating than standing in line with a bunch of stamps,” and four other participants agreed.

## 4. Discussion

Interviews with SSI recipients newly receiving SNAP benefits illustrated that SNAP had positive benefits on access to nutritious food while reducing the amount of labor necessary to access food. SNAP allowed beneficiaries to divert income to the purchase of basic necessities that they would have gone without. The addition of SNAP to a tight SSI budget provided some relief to the mental stress of poverty.

Even with the addition of SNAP benefits, many participants continued to experience barriers to healthy dietary intake. Food budgets for many were still extremely tight, especially because SNAP benefit levels are not adjusted to county-level prices [29]. Many participants struggled to make meals from low-cost ingredients rather than purchasing prepared foods. Others relied on other benefits, including appropriate housing and home care resources, to be able to make use of their SNAP benefits. Thus, assistance with grocery shopping and meal preparation as well as the space to prepare food are important components of translating SNAP participation into improved nutrition for older or disabled adults. SNAP participants with mobility impairments are particularly vulnerable to loss of purchasing power in SNAP benefits [30]. Our findings support a growing number of pilot and COVID-related programs allowing SNAP redemption online, which can expand access to affordable grocery stores while reducing physical access barriers [31,32]. These findings help explain the broader study’s quantitative results which demonstrated that participants did not experience a significant change in their dietary intake or other health outcomes after the policy change, although they were more likely to report improved general health status and less cost-related medication nonadherence at follow up compared to baseline. 

Marginally housed people experience food insecurity at very high rates. Bowen and colleagues found a 75% food insecurity rate among SRO tenants in Chicago, despite high SNAP participation rates [33]. Brown and colleagues found an 84% food insecurity rate among SRO tenants in San Francisco [34]. The authors of both articles hypothesized that lack of food preparation and food storage space may be important contributing factors to these high rates of food insecurity and called for qualitative research exploring how SRO tenants’ housing restraints impact food security and how they navigate food assistance programs. Findings from this study affirm the lack of kitchen and pantry space in these types of housing as a barrier to food access. Given the high rate of housing insecurity among SNAP recipients, addressing these housing barriers should be a priority for housing policymakers and program administrators. 

Participants who concerned themselves with SNAP fraud appeared to be internalizing widespread stigma and misperceptions that fraud is a significant issue. Actual fraud rates for SNAP are low, less than 1% [35]. However, the prominence of concerns about fraud among new SNAP recipients highlights their awareness of the stigma of SNAP reliance as well as, potentially, the increase in welfare retrenchment discourse in recent years. A recent study found that although SNAP was associated with improved physical health outcomes, its use was also correlated with negative self-attitudes and depressive symptoms [36]. Taken together, the authors suggest that the stigmatizing effects of SNAP may offset the otherwise protective factors SNAP could have on participant mental health. Thus, the fact that stigma emerged as a major theme despite its absence from our interview questions highlights stigma’s primacy in the overall experience of SNAP receipt. It further suggests that policies aimed at reducing the stigma of SNAP enrollment for older and disabled adults may be beneficial both for optimizing the health-protective effects of SNAP and for increasing eligible participant enrollment. One such policy approach could be automatic enrollment into SNAP for older adults or adults with disabilities meeting certain income guidelines [37]. 

Our study has several limitations. First, our sample was drawn from a single urban setting and was limited to participants proficient in the English language. Second, our sample was restricted to participants in the parent study who (1) agreed to be contacted for in-depth interviews and (2) participated in both the baseline and follow up study activities. Thus, there could be some selection bias amongst our sample. While our findings may have limited generalizability, qualitative research typically does not seek generalizability, rather to further understanding of phenomena. Our study’s sample focused on the experiences of older adults in a large metropolitan area. Future studies could assess the applicability of these findings amongst younger SSI beneficiaries and those in suburban or rural areas. Our sample received SNAP benefits for an average of approximately six months. Future studies may benefit from a longer intervention duration of SNAP benefit receipt in order to observe the effects of longer-term use of SNAP. Further, conducting interviews with SSI beneficiaries who did not enroll in SNAP could increase understanding of barriers to enrollment for this population.

## 5. Conclusions

New SNAP eligibility and enrollment overall improved the lives of our study’s participants by increasing their ability to buy healthy foods of their choice, contributing to overall budgets, easing mental distress resulting from poverty, and reducing labor spent in accessing food. The depictions of the daily lives of these older, disabled adults highlighted the need for adequate housing and assistance in completing tasks of daily living in order to make optimal use of SNAP benefits. Together, the impacts of and barriers to effective use of SNAP benefits gleaned from this study deepens our understanding of individual- and neighborhood-level factors driving poor health among people experiencing food insecurity and SNAP recipients. Finally, the degree to which welfare-related stigma permeated participants’ discussions of SNAP benefits, in context with recent literature on the impacts of this stigma, motivates further assessment of the degree to which stigma may prevent SNAP participation among older adults and investigation into programmatic and policy interventions to reduce it.

## Figures and Tables

**Table 1 nutrients-13-04362-t001:** Characteristics of a cohort of 20 older adult SSI recipients in California who participated in in-depth interviews on the impacts of SNAP after recent benefit enrollment.

	Number *n*
**Gender**	
Female	9
Male	11
**Age**	
58–64	8
66–69	6
70–77	6
**Race/ethnicity** ^a^	
American Indian/Alaskan Native	1
Asian	2
Black/African American	9
Hispanic/Latino	3
White	6
Other	1
**Highest educational level attained**	
Some high school	4
High school/GED	3
Some college/technical school	12
College graduate	1
**Household size**	
Just myself	18
Myself and one other person	2
**Housing type**	
Federal housing subsidy (Section 8)	5
Senior housing	1
Single Room Occupancy/Hotel/Motel	8
Supportive, low income	5
Other	1
**Household annual income**	
Less than $5001	3
$5000–$10,000	8
$10,001–$15,000	9
**Monthly SNAP benefit amount** ^b^	
$20–$38	4
$43–$54	5
$61–$90	6
$137–$190	5
**Participation in at least one other food resource, baseline**	
Yes	14
No	6
**Additional food resources used, baseline** ^c^	
Free groceries	11
Free dining room or soup kitchen	5
Home delivered meals	5
Fruit and vegetable vouchers	1
Other	1
**Self-reported health, baseline**	
Excellent	2
Very Good	6
Good	4
Fair	7
Poor	1

^a^ N > 20 reflects the possibility of selecting multiple races or ethnicities by one participant. ^b^ The average monthly SNAP benefit amount was $80.55 (SD = 54). ^c^ N > 20 reflects participants indicating use of multiple food resources.

## Data Availability

The data that support the findings of this study are available on request from the corresponding author, K.S. The data are not publicly available due to their containing information that could compromise the privacy of research participants.

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
