# Peer review of "“Now I Feel a Little Bit More Secure”: The Impact of SNAP Enrollment on Older Adult SSI Recipients"

_nutrients, 2021, doi:10.3390/nu13124362_

Round 1
Reviewer 1 Report
The paper entitled “Now I Feel a Little Bit More Secure:” The Impact of SNAP Enrollment on Older Adult SSI Recipients deals with important question from social point of view.
Additional comments:
- the group of participants is quite small (even for qualitative analysis)
- table 1: data should be presented only in numbers and not percentages (due to low number of subjects in subgroups)
- how long were the subjects beneficiaries of SNAP? It was mentioned in inclusion criteria "new SNAP enrollment since the policy change". Was the time long enough to perceive the impact of SNAP by subjects?
- the background of social programs (and some terms) are hard to understand/imagine for a reader not living in US
- there is lack of appendix to see a semi-structured interview guide
Author Response
Response to Reviewer 1 Comments
Thank you for your comments, they were very helpful.
Point 1: the group of participants is quite small (even for qualitative analysis)
Response 1: Since qualitative research sample sizes are typically determined by data saturation rather than number of participants, we do not believe our sample was too small for the project at hand and found that we did reach the point where additional data ceased to generate additional codes. We do acknowledge the limits to generalizability of our findings in our discussion section.
Point 2: table 1: data should be presented only in numbers and not percentages (due to low number of subjects in subgroups)
Response 2: The percentage column has been deleted.
Point 3: how long were the subjects beneficiaries of SNAP? It was mentioned in inclusion criteria "new SNAP enrollment since the policy change". Was the time long enough to perceive the impact of SNAP by subjects?
Response 3: The CalFresh policy change occurred in June 2019, and participants were interviewed in the Winter/Spring of 2019/2020. The duration of SNAP benefits were approximately 6 months on average. The interviewers asked how long they had been on SNAP towards the beginning of the interview. For some, one month of SNAP helped them make payments towards other bills immediately. For others, one month of SNAP did not make a difference due to the low benefit amount, so the impact would not likely change with longer duration of benefits. Additionally, we asked about the SNAP enrollment process, so having participants who had recently enrolled helped us delve into their perspective on that aspect. We agree that for long term changes like health outcomes, our timeline would not assess such impacts. However, we were able to tailor the question on impact to each participant based on their unique circumstances. In addition, we have added a sentence on SNAP benefit duration to the limitations section.
Point 4: the background of social programs (and some terms) are hard to understand/imagine for a reader not living in US
Response 4: We reviewed each term in the background section to make sure it was clearly defined for an international audience and added a definitional parenthetical for Medicare/Medicaid.
Point 5: there is lack of appendix to see a semi-structured interview guide
Response 5: We have added an appendix with the interview guide.
Reviewer 2 Report
This is a worthwhile topic and overall the paper are important.
Usually grounded theory based research includes a table with the themes and the number of participant who fit each theme . It would be useful to have this data since the "child" theme have been really condense and actually I have the sense that the authors "found what they were looking " so to speak. For example , how come there is no information of the cultural background of the 22 participants ? Yet when the food significance is discussed we have a quotation from a Portugese participant. Moreover , the theme of food as a cultural value was not found in the first interviews but rather suggested by the researchers in second stage?
One other point that perhaps needs to be considered is the limitation that arise from such a low number of participants when it comes to talking about stigma .
Author Response
Response to Reviewer 2:
Thank you for your thoughtful comments.
Point 1: Usually grounded theory based research includes a table with the themes and the number of participant who fit each theme . It would be useful to have this data since the "child" theme have been really condense and actually I have the sense that the authors "found what they were looking " so to speak. For example , how come there is no information of the cultural background of the 22 participants ? Yet when the food significance is discussed we have a quotation from a Portugese participant. Moreover , the theme of food as a cultural value was not found in the first interviews but rather suggested by the researchers in second stage?
Response 1: We did not set out searching for evidence of stigma or cultural themes related to food in our qualitative data analysis. These themes truly emerged as described in our methods, through individual and team coding for semantic and latent codes. If you have any suggestions as to how to increase trustworthiness that our methods are truly as written without predetermined findings, we would welcome the feedback.
Also of note, we did not use grounded theory, instead we used thematic analysis as our approach to qualitative data analysis (though inclusion of a table with the number of participants who fit each theme is not typical for grounded theory either, it is done on occasion to appeal to quantitative researchers who may be reviewing qualitative analysis.) In response to your note on the condensing of child and parent codes, we have added a table in the appendix.
We did identify the cultural component of food in the initial coding process. The second paragraph in the “Role of Food” section provides an example of how this code emerged organically again during informal discussion during our member check groups.
Point 2: One other point that perhaps needs to be considered is the limitation that arise from such a low number of participants when it comes to talking about stigma .
Response 2: We do acknowledge the limits to generalizability of our findings due to the sample size in our discussion section. While we did not originally seek out to study stigma, stigma was a clear theme in participant responses - even though they were never directly asked about it. Thus, it is included in our findings in our efforts to report truthfully the findings that emerged. Further, this theme is reflected in literature from other studies about the experience of receiving SNAP benefits.
Reviewer 3 Report
Sound presentation
Author Response
Thank you for your review of this manuscript.
Reviewer 4 Report
The present manuscript evaluates the impact of SNAP program. The sample is
adequate given the fact that it was an in depth interview process. The limitations I had in mind are clearly presented by the authors.
This is an interesting work.
Author Response
Thank you for your thoughts and your review of this manuscript.